# Risk Factors and Outcomes of Patients Colonized with KPC and NDM Carbapenemase-Producing Enterobacterales

**DOI:** 10.3390/antibiotics13050427

**Published:** 2024-05-08

**Authors:** Lisa Saidel-Odes, Orli Sagi, Shani Troib, Hannah Leeman, Ronit Nativ, Tal Schlaeffer-Yosef, Hovav Azulay, Lior Nesher, Abraham Borer

**Affiliations:** 1Infection Control and Hospital Epidemiology Unit, Soroka University Medical Center, Beer Sheba 84101, Israel; ronitna@clalit.org.il (R.N.); abrahamb@clalit.org.il (A.B.); 2The Faculty of Health Sciences, Ben-Gurion University of the Negev, Beer Sheba 84105, Israel; leemanh@post.bgu.ac.il (H.L.);; 3Medical Microbiology Laboratory, Soroka University Medical Center, Beer Sheba 84101, Israel; 4Infectious Disease Institute, Soroka University Medical Center, Beer Sheba 84101, Israel

**Keywords:** antibiotic stewardship, beta-lactamase NDM, infection control, multidrug resistance

## Abstract

Carbapenemase-producing enterobacterales (CPE) poses an increasing threat in hospitals worldwide. Recently, the prevalence of different carbapenemases conferring carbapenem resistance in enterobacterales changed in our country, including an increase in New Delhi Metallo-beta-lactamase (NDM)-CPE. We conducted a comparative historical study of adult patients colonized with *Klebsiella pneumoniae* carbapenemase (KPC)-CPE (July 2016 to June 2018, a historical cohort) vs. NDM-CPE (July 2016 to January 2023). We identified patients retrospectively through the microbiology laboratory and reviewed their files, extracting demographics, underlying diseases, Charlson Comorbidity Index (CCI) scores, treatments, and outcomes. This study included 228 consecutive patients from whom a CPE rectal swab screening was obtained: 136 NDM-CPE positive and 92 KPC-CPE positive. NDM-CPE-colonized patients had a shorter hospitalization length and a significantly lower 30-day post-discharge mortality rate (*p* = 0.002) than KPC-CPE-colonized patients. Based on multivariate regression, independent risk factors predicting CPE-NDM colonization included admission from home and CCI < 4 (*p* < 0.001, *p* = 0.037, respectively). The increase in NDM-CPE prevalence necessitates a modified CPE screening strategy upon hospital admission tailored to the changing local CPE epidemiology. In our region, the screening of younger patients residing at home with fewer comorbidities should be considered, regardless of a prior community healthcare contact or hospital admission.

## 1. Introduction

Carbapenemase-producing enterobacterales (CPE) continues to be one of the significant challenges in hospitals and other healthcare facilities worldwide, posing a major public health threat with considerable morbidity and mortality [1,2,3]. CPE is a concern in healthcare settings due to its ability to produce beta-lactamase enzymes that can hydrolyze most beta-lactam antibiotics, making it resistant to many antimicrobial agents, including carbapenems, which are a last resort for treating severe infections. Carbapenemases are classified into Ambler class A, class B, and class D. Among them, the most broadly spread carbapenemases are *Klebsiella pneumoniae* carbapenemase (KPC), New Delhi metallo-beta-lactamase (NDM), oxacillinase (OXA)-48, imipenemase (IMP), and Verona integron-encoded metallo-beta-lactamase (VIM) [4]. KPC is a class A β-lactamase that has the capacity to hydrolyze penicillins, cephalosporins, and carbapenems. NDM is a class B β-lactamase capable of hydrolyzing penicillins, cephalosporins, and carbapenems, but unlike KPC, it does not hydrolyze aztreonam [5]. CPE is readily transmissible to other Gram-negative organisms because the genes encoding carbapenemases are located on mobile genetic elements such as plasmids, transposons, and insertion sequences, which can spread easily from patient to patient in healthcare settings [1,6]. Studies have demonstrated that CPE colonization can persist for over 12 months once a patient has acquired it [7]. CPE infections are highly complicated to manage due to the extremely limited treatment options and are associated with high mortality, which can reach upward of 40% when patients suffer from an invasive CPE infection [5].

Carbapenem-resistant enterobacterales with NDM-type carbapenemases continued to spread globally over the years and are currently isolated in clinical settings worldwide, from Europe to Asia, the United States, South America, and Africa [8]. In the past few years, NDM-producing enterobacteriaceae appear to be disseminating from South Asia [5]. NDM variants, NDM-1 and NDM-5, are the most common [9]. There is a substantial variability at the continental, national, regional, and even center-to-center levels. Awareness of the prevalence and incidence of CPE-NDM is crucial in preventing its spread.

Historically, KPC-producing *Klebsiella pneumoniae* was the first to emerge and spread globally and became endemic in the United States, Israel, Greece, and Italy [5]. The predominant carbapenemase during the initial years of CPE spread in our nation’s hospitals, starting in 2006, was KPC, harbored by *Klebsiella pneumoniae*, accounting for over 94% of CPE cases [10]. However, in the past five years, the epidemiology of CPE in terms of pathogens and resistance mechanisms has changed. The resistance mechanism of KPC-CPE has decreased to less than 30% of all CPE isolates, while we observed an increase in NDM-CPE, which has become widespread and accounts for 29–33% of new CPE cases [11].

NDM-positive strains cause various infections, reported in association with high mortality rates [12]. These strains are found worldwide, representing a significant clinical management and infection control challenge. Patients colonized with NDM-CPE appear to represent a different patient population than those colonized with KPC-CPE. The early identification of patients harboring NDM-CPE is essential for promptly implementing cohort-nursing practices and minimizing the spread and subsequent related morbidity and mortality. To implement this strategy, we need to determine the best measures on which to screen to identify patients colonized with NDM-CPE.

Our institution implemented a multifaceted strategy to prevent KPC-CPE dissemination, including a flagging system in the emergency room that identifies high-risk groups based on epidemiology and clinical risk factors for KPC-CPE colonization. We obtained rectal swabs for CPE from high-risk groups and pre-emptively isolated patients until the culture results were available [13]. Patients found positive for CPE colonization (with or without infection) were moved to a CPE cohort in a dedicated unit in one of our Internal Medicine wards.

In this study, we aim to understand the epidemiology better and develop effective screening strategies, thus identifying and managing the spread of NDM-CPE by comparing it to previously studied KPC-CPE colonized patients [14]. We are unaware of other studies that have compared risk factors and outcomes in NDM-CPE versus KPC-CPE colonized and non-infected patients.

## 2. Results

We identified and included 228 consecutively hospitalized adult patients with a rectal swab positive for CPE in this study: 92 positives for KPC-CPE obtained between July 2016 and June 2018 (a historical cohort [14]), and 136 positives for NDM-CPE obtained between July 2016 and January 2023.

The patient’s demographic and epidemiological characteristics are presented in Table 1. We compared patients colonized with NDM-CPE to those colonized with KPC-CPE. CPE- NDM-colonized patients were younger, 60.7 ± 19.56 vs. 67.2 ± 18.78 (*p* = 0.013); more likely to be admitted from home (as opposed to a nursing care facility), 80.9% vs. 31.5% (*p* = 0.001); and had fewer comorbidities with a lower CCI, 4.18 ± 3.11 vs. 5.87 ± 3.28 (*p* < 0.001). KPC-CPE-colonized patients had a higher rate of ischemic heart disease, peripheral vascular disease, liver disease, and dementia (Appendix A). Of the CPE-colonized patients admitted from home, 79 (71.8%) of NDM-CPE-colonized patients vs. 18 (62%) of KPC-CPE-colonized patents had a previous hospitalization in the past six months (*p* = 0.364).

NDM-CPE-colonized patients were more likely to be admitted to a surgical department than KPC-CPE-colonized patients, 27.9% vs. 13% (*p* = 0.007), and less likely to be admitted to an Internal Medicine department, 61% vs. 81.5% (*p* < 0.001).

The patient’s clinical characteristics during hospitalization and outcomes are shown in Table 2. Comparing patients colonized with NDM-CPE to those colonized with KPC-CPE, they were less likely to have a urinary catheter, 25.7% vs. 47.8% (*p* = 0.001), or a decubitus ulcer, 16.2% vs. 54.9% (*p* < 0.001). Their hospital length of stay tended to be shorter. The in-hospital mortality rate was similar in both groups, but the 30-day post-discharge mortality rate was significantly lower in NDM-CPE-colonized patients vs. KPC-CPE-colonized patients, 4.2% vs. 18.4% (*p* = 0.002). The NDM-CPE gene was mainly found with *Escherichia coli* (67.6%), whereas the KPC-CPE gene is more commonly found with *Klebsiella* sp. (68.5%) (*p* < 0.001, for both). Utilizing univariate analysis, the risk factors predicting CPE-NDM colonization included age < 65 y/o (*p* = 0.041, OR:1.82 95% CI 1.05–3.14), admission from home (*p* < 0.001, OR:9.19 95% CI 4.98–16.71), and CCI < 4 (*p* = 0.004, OR:2.24 95% CI 1.29–3.89).

Based on multivariate regression (Table 3), the independent risk factors predicting CPE-NDM colonization included admission from home (*p* < 0.001, OR: 0.15, 95% CI 0.07–0.32) having less comorbidities with CCI< 4 (*p* = 0.037, OR:0.39, 95% CI 0.16–0.94), and the absence of a urinary catheter and decubitus ulcers.

## 3. Discussion

The epidemiology of CPE in Israeli hospitals has changed in terms of pathogens and resistance mechanisms in the past few years. While the predominance of KPC-producing *Klebsiella pneumoniae* has declined, NDM-producing *Escherichia coli* and *Enterobacter* sp. have risen significantly in recent years [11,12]. Hospitals apply a risk-based screening strategy upon patient admission to identify the carriers of CPE and implement infection control measures to contain the spread of CPE [4,15,16]. Deciding which patients to screen is based on risk factors for CPE colonization. However, the predisposing factors for colonization by NDM-CPE remain largely under-investigated [12]. The extent of NDM-CPE spread in the community remains unclear, although there are reports of NDM-positive strains detected in healthy individuals [12].

Our current strategy for CPE screening on patient admission is based on risk factors for KPC-CPE colonization. In a previous study, we found that older age, nursing home residency, prior antibiotic treatment, and the presence of a decubitus ulcer were independent risk factors predicting KPC-CPE colonization [14]. Other studies looked at risk factors for the acquisition of CPE; however, they did not subdivide them according to the Ambler class [4,17]. These studies found that patients with long and frequent hospital admissions, patients receiving antimicrobial treatment for ten days or longer, and patients receiving hemodialysis were more likely to be colonized with CPE.

This study found that the risk factors for NDM-CPE colonization differ from those for KPC-CPE colonization. NDM-CPE-colonized patients were younger, with fewer comorbidities, had significantly less decubitus ulcers or a urinary catheter, and mainly resided at home and not in nursing home facilities.

Patients’ outcomes differed between the two groups. NDM-CPE-colonized patients had a shorter hospitalization duration compared to KPC-CPE-colonized patients. Although there was no difference in hospital mortality between the two groups, there was a significantly lower 30-day post-discharge mortality among NDM-CPE-colonized patients. This difference in outcomes could be explained by these patients’ younger ages and lower rates of comorbidities. Similar findings were shown in a study performed by Seo et al. [18]. This was a retrospective cohort study at a 2700-bed tertiary referral hospital in Seoul, South Korea, during 2010–2019, which compared the clinical outcomes of patients with CPE-KPC and CPE-NDM colonization and/or infection. During the study period, they identified 859 patients ≥16 y/o colonized and/or infected with CPE at baseline; 475 (55%) had KPC, of which 122 had CPE-KPC infection, and 384 (45%) had NDM, of which 89 had CPE-NDM infection. The 30-day mortality rate after the first isolation of CPE-KPC vs. CPE-NDM was significantly higher in the KPC group than in the NDM group (17% vs. 9%, *p* < 0.001). Although our patient cohort were colonized but not infected with CPE, our 30-day post-discharge mortality rate for CPE-KPC versus CPE-NDM was similar to the 30-day mortality rate after the first isolation of CPE in the study by Seo et al. [18].

We found a significant difference in species distribution for the different carbapenemase genes (*p* < 0.001); the most common species–gene combination was *Klebsiella pneumoniae* among KPC-CPE (68.5%) and *Escherichia coli* among NDM-CPE (67.6%). Other studies found similar findings regarding KPC-CPE [4,12], though NDM-CPE was more common with *Enterobacter cloacae* complex in a study by Assis et al. [12]. The SMART Global Surveillance Program collected strains of NDM-CPE from 55 countries, among which *Klebsiella pneumoniae* was the most common species (>58%), followed by *Escherichia coli* (17%) and *Enterobacter cloacae* complex (13%) [19]. A study in the United Kingdom that analyzed their first 250 cases of NDM-KPC showed similar results; the predominant hosts were *Klebsiella pneumoniae* (55%) and *Escherichia coli* (25%) [15]. Notably, most of their samples were clinical samples, and only a minority (14%) were screening swabs. In an extended survey in the United Kingdom, India, Pakistan, and Bangladesh in 2008–2009, NDM was found in many isolates, predominantly *Klebsiella pneumoniae* and *Escherichia coli* [20]. NDM-producing enterobacterales bacteria during 2019–2020 accounted for >25% of all CPEs found in Switzerland; in a study by Findlay et al., more than half of the isolates (82/141, 58.2%) were obtained from screening swab samples (fecal, rectal, and non-rectal). Most isolates were either *Klebsiella pneumoniae* (41.8%) or *Escherichia coli* (36.9%) [21]. A study in a university hospital in Madrid, Spain, from March 2014 to March 2016 collected 15,556 rectal swabs from 8209 patients admitted in two surgical and two medical wards. They identified 198 CPE isolates. The most frequent carbapenemase was OXA-48 (64.1%); NDM-1 was found in 5.3% of these isolates [22]. A systemic review of the epidemiology, risk factors, and clinical outcomes of carbapenem-resistant enterobacterales in Africa found that the most frequently detected carbapenemases were NDM (43.1%) and OXA-48-like (42.9%). The most common bacterial isolates were *Klebsiella pneumoniae* and *Escherichia coli*. [23]. A study conducted in China by Li et al. collected 685 fecal individual samples: 544 from five hospitals in four distinct provinces and 141 from healthy individuals. Of these 685 fecal samples, 66 carbapenem-resistant enterobacterales strains were isolated, representing a carriage rate of 9.6%, of whom 97% were CPE-NDM. Of the 141 fecal samples from healthy individuals, six carbapenem-resistant *Enterobacterales* strains were identified, with a carriage rate of 4.26%, all were CPE-NDM. *Escherichia coli* was the most prevalent bacteria (57.6%) followed by *Klebsiella* (15.15%), *Citrobacter* (13.6%), and *Enterobacter* (12.1%) [24]. A study conducted in India by Arum et al. screened 1000 stool samples for CPE from healthy individuals from three cities. Fecal carriage for CPE among healthy individuals was 6.1%; 28/61 showed blaNDM-1 and 33/61 blaOXA48 [25]. In our region, the CPE carriage rate (for all carbapenemases) in hospitalized patients based on our screening rectal cultures is 1.3%; this is substantially lower than the rate described in China and India, though a cause for concern.

Our study has some limitations. Firstly, this is a retrospective study in a single medical center serving a unique population, which may not be generalizable in all aspects to other medical centers. Secondly, there are no available data regarding the prevalence of NDM-CPE in the community in Southern Israel. Patients are screened on hospital admission according to KPC-CPE screening criteria, and there is no follow-up after discharge.

## 4. Materials and Methods

### 4.1. Study Design

We performed a comparative historical study of NDM-CPE and KPC-CPE colonization in adult patients hospitalized at Soroka University Medical Center (1100 beds, approximately 80,000 admissions annually). Our patient population included residents of the Negev (Southern Israel) comprised patients of Jewish and Bedouin-Arab ethnicity (75% and 25%, respectively).

### 4.2. Study Population

From July 2016 to January 2023, all rectal culture screenings for CPE collected from the bacteriology laboratory were reviewed, and only one NDM-CPE screening rectal culture (first culture) per patient was included in this study. Rectal cultures from hospitalized patients aged ≥ 18 y/o were included for analysis. Screening rectal cultures for CPE are performed in our hospital on patient admission and during hospitalization according to local and national guidelines. Screening rectal cultures on admission are performed for the following indications: all bedridden patients, patients with a prior rectal swab positive for CPE in the past three years, patients admitted from other hospitals and nursing homes, patients that were hospitalized in other healthcare facilities in the past six months, patients readmitted to our hospital within six months of a previous hospitalization discharge, and all patients admitted to an intensive care unit. Screening rectal cultures during hospitalization are performed for the following indications: patients transferred between wards in our hospital upon admission to the new ward, all intensive care unit patients negative for CPE upon admission undergo a weekly screening rectal culture until discharge from the unit, and, in all hospital wards, any contact of a patient who was found positive for CPE during hospitalization was screened (we have very few single patients rooms; therefore, contacts are patients who shared a room with a newly discovered carrier). We compared patients colonized with NDM-CPE with a historical cohort of previously studied KPC-CPE-colonized patients [14]. CPE screening criteria throughout the entire study period (including the historical cohort) did not change.

### 4.3. Measures

We collected data using a pre-designed structured questionnaire covering demographic background, including age, gender, ethnicity, smoking status, place of residence (home vs. nursing care facility), the ward where the patient was hospitalized, the Charlson Comorbidity Index (CCI), and underlying diseases. We also included information on prior antibiotics and immunosuppressive therapy when it occurred ≤ 3 months preceding index admission. Previous hospital admissions and nursing home residency six months prior to admission were identified and included in the analysis. Furthermore, we collected data on intensive care unit (ICU) admission, mechanical ventilation, a permanent urinary catheter, a central line, a nasogastric feeding tube, a decubitus ulcer, bedridden, length of stay, hospital mortality rate, and the 30-day crude mortality rate. Bacteriology data included the bacteria in which meropenem resistance was detected and the carbapenemase present (NDM and KPC).

### 4.4. Microbiological Analysis

All rectal swabs were inoculated onto a CHROMagar mSuperCARBA (HyLabs, Rehovot, Israel) to isolate and detect suspected carbapenemase-resistant enterobacteriaceae (CRE). Isolates were identified by VITEK-MS (bioMérieux, Craponne, France) and tested against meropenem using disc diffusion (Oxoid, Basingstoke, UK) and E-test methods (bioMérieux). We used the Clinical and Laboratory Standard Institute (CLSI) standards to define resistance. For meropenem-resistant strains, carbapenemase production was confirmed using GeneXpert^®^ Carba-R assay (Cepheid, Sunnyvale, CA, USA) to detect and differentiate KPC, NDM, VIM, OXA-48 and IMP. CPE was determined in the case of a meropenem-resistant strain and GeneXpert positive results.

### 4.5. Statistical Analysis

We analyzed the data using SPSS version 26.0 (Chicago, IL, USA: SPSS Inc.) and R version 3.5.3 (The R Foundation for Statistical Computing, Vienna, Austria: www.r-project.org). We analyzed categorical variables using a Chi-square assay or Fisher’s extract test, while continuous variables were analyzed using an independent samples *t*-test or a Mann–Whitney U test. Variables that were found to be significantly associated with the different outcomes (*p*-value < 0.05) during univariate analyses were then gradually added to stepwise selection multivariable models. Odds ratios (ORs)/hazard ratios (HRs) and 95% confidence intervals (CIs) were calculated to evaluate the strength of each association. The association between the significant independent variables and the dichotomous outcomes were analyzed using multivariable logistic regressions, while the association between the different variables and mortality during hospitalization was examined using Cox regression.

## 5. Conclusions

As NDM-CPE prevalence increases, the CPE screening strategy upon hospital admission should be tailored to the changing local CPE epidemiology. This strategy should consider risk factors for NDM-CPE colonization in addition to, and may differ from, known risk factors for KPC-CPE colonization. In our region, in addition to screening patients with recurrent admissions and patients admitted from nursing homes, one should also screen younger patients residing at home with fewer comorbidities and who are admitted to surgical or medical wards. Utilizing a risk-based targeted screening strategy will allow a more effective strategy for the early detection of CPE carriage, allowing for promptly implementing appropriate infection control measures before transmission occurs within hospitals, thus mitigating the in-hospital spread of CPE.

## Figures and Tables

**Table 1 antibiotics-13-00427-t001:** Demographic and epidemiological characteristics of the patient population: KPC-CPE-colonized patients vs. NDM-CPE-colonized patients.

Variable		KPC-CPE n = 92	NDM-CPE n = 136	*p*-Value	OR [95% CI]
Age	mean ± SD	67.2 ± 18.78	60.7 ± 19.56	0.013	
	median (range)	70.5 (19–94)	65.5 (19–98)		
Male sex		47 (51.1%)	83 (61%)		
Origin	Home	29 (31.5%)	110 (80.9%)	0.001	0.108 [0.05–0.2]
	Nursing home	42 (45.7%)	2 (1.5%)	<0.001	56.23 [13.13–241.2]
	Other	21 (22.8%)	24 (17.6%)	0.336	
Department	Internal medicine	75 (81.5%)	83 (61%)	<0.001	2.81 [1.5–5.28]
	Surgery	12 (13%)	38 (27.9%)	0.007	0.38 [0.19–0.79]
	ICU	3 (3.3%)	8 (5.9%)	0.364	
	Other	2 (2.2%)	7 (5.1%)	0.258	
Charlson Index	mean ± SD	5.87 ± 3.28	4.18 ± 3.11	<0.001	
	median (range)	6 (0–16)	4 (0–13)	<0.001	
Smoker		21 (22.8%)	36 (26.5%)	0.640	
Antibiotic treatment three months prior		55 (59.8%)	93 (68.4%)	0.204	
Steroid treatment three months prior		16 (17.4%)	29 (21.3%)	0.502	
Recurrent admission within six months		58 (63%)	96 (70.6%)	0.251	

**Table 2 antibiotics-13-00427-t002:** Clinical characteristics during hospitalization and outcome of CPE patient population: KPC-CPE-colonized patients vs. NDM-CPE-colonized patients.

Variable			KPC-CPE n = 92	NDM-CPE n = 136	*p*-Value	OR [95% CI]
Mechanical ventilation			14 (15.2%)	18 (13.2%)	0.701	
Hemodialysis			6 (6.5%)	11 (8.1%)	0.799	
Urinary catheter			44 (47.8%)	35 (25.7%)	0.001	0.38 [0.22–0.66]
Decubitus ulcer			50 (54.9%)	22 (16.2%)	<0.001	0.16 [0.09–0.29]
Nosocomial CPE colonization			29 (31.5%)	55 (40.7%)	0.165	
Bacteria	*Escherichia coli*		6 (6.5%)	92 (67.6%)	<0.001	0.03 [0.001–0.08]
	*Klebsiella* sp.		63 (68.5%)	17 (12.5%)	<0.001	15.2 [7.76–29.8]
	Enterobacter		15 (16.3%)	24 (17.6%)	0.791	
	Other		8 (8.7%)	3 (2.3%)	0.024	4.2 [1.1–16.3]
Length of stay, days	Mean ± SD		13.29 ± 17.16	10.1 ± 10.59	0.086	
	Median (range)		6 (2–93)	6 (2–59)		
In-hospital mortality			11 (12%)	15 (11%)	0.913	
30-day post discharge mortality			16 (18.4%)	5 (4.2%)	0.002	0.19 [0.07–0.55]
Discharged to		Home	25 (27.2%)	103 (75.7%)	<0.001	0.11 [0.06–0.29]
		Nursing home	56 (60.9%)	13 (9.6%)	<0.001	14.7 [7.24–29.9]

**Table 3 antibiotics-13-00427-t003:** Multivariate regression of independent risk factors predicting NDM-CPE colonization.

Variable	*p*-Value	OR	95% CI	
			Lower	Upper
Age < 65	0.620	1.254	0.512	3.068
Admission from home	0.000	6.536	3.105	13.888
Charlson Index < 4	0.037	2.525	1.059	6.024
No urinary catheter	0.034	2.164	1.058	4.426
No decubitus ulcer	0.042	2.250	1.031	4.972

## Data Availability

The datasets generated and analyzed during the current study are not publicly available but are available from the corresponding author upon reasonable request.

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
