# Peer review of "Risk Factors and Outcomes of Patients Colonized with KPC and NDM Carbapenemase-Producing Enterobacterales"

_antibiotics, 2024, doi:10.3390/antibiotics13050427_

Round 1

Reviewer 1 Report

Comments and Suggestions for Authors

The article under review is extremely interesting and well-developed. Its introduction provides a concise and comprehensive overview of the topic, offering the reader a clear understanding of the context. However, the positioning of the materials and methods could be improved by placing them between the introduction and the results to ensure a better logical structure and understanding of the research flow.

Furthermore, it would be advantageous to further elaborate on the study design, providing more details on the methodologies employed. The results have been described clearly and comprehensively, both through the text and via tables, facilitating data comprehension and interpretation.

Lastly, the discussion is particularly engaging and well-supported, especially concerning the comparison between the two strains, which is enriched by critical analysis of the existing literature.

Author Response

Remark 1: The article under review is extremely interesting and well-developed. Its introduction provides a concise and comprehensive overview of the topic, offering the reader a clear understanding of the context. However, the positioning of the materials and methods could be improved by placing them between the introduction and the results to ensure a better logical structure and understanding of the research flow.

Answer: Thank you for this comment, with regards to the positioning of the materials and methods section, we followed the journal's instruction for authors, manuscript preparation section which stated the following order "Research manuscript sections: Introduction, Results, Discussion, Materials and Methods, Conclusions ".

Remark 2: Furthermore, it would be advantageous to further elaborate on the study design, providing more details on the methodologies employed.

Answer: We elaborated on the study design. Page 5, lines 186-187. Page 6 lines 204-205.

Remark 3: The results have been described clearly and comprehensively, both through the text and via tables, facilitating data comprehension and interpretation.

Answer: Thank you

Remark 4: Lastly, the discussion is particularly engaging and well-supported, especially concerning the comparison between the two strains, which is enriched by critical analysis of the existing literature.

Answer: Thank you

Reviewer 2 Report

Comments and Suggestions for Authors

In the manuscript “Risk Factors and Outcomes of Patients Colonized with KPC and NDM Carbapenemase-Producing Enterobacterales”, the researchers compared the patients’ characteristics between the groups tested positive for KPC-CPE vs NDM-CPE. The researchers found characteristics such as admission origin and Charlson Comorbidity Index (CCI) were associated with the groups. While the research question is interesting, major improvements are needed.

Major comments:

1.    The univariate and multivariate regression analyses can only demonstrate associations between patients’ characteristics and KPC-CPE vs NDM-CPE colonization. AUCs or F1 scores on a test dataset are needed to demonstrate the predictive values of these characteristics. This is a major concern as predicting NDM-CPE patients is one conclusion from the paper.

2.    How do the characteristics distinguish between NDM-CPE vs non CPE patients?

3.    Please clarify on which comorbidity these patients had and whether any of them were associated with the groups. Knowing the specific co-morbidities associated with NDM-CPE can be more informative than CCI.

4.    Since the characteristics were added stepwise to the multivariate regression model, what was the order of adding the characteristics? How were the different multivariate regression models compared and which model was determined as the final model?

5.    From Table 3, all the variables were associated with the groups (p<0.05) except Age < 65, unless a different p value threshold was used. If a different threshold was used, please clarify, if not, why the other variables (urinary catheter and decubitus ulcer) were not mentioned and discussed?

Minor comments:

1.  I would suggest changing “sig.” in Table 3 to “p-value”, same as in Table 1&2.

2.  Although explained in the results section, I would suggest to also add footnotes in the tables to clarify the direction of the odds ratios between the groups.

Author Response

Answers to Reviewer #2

Reviewer 2: In the manuscript “Risk Factors and Outcomes of Patients Colonized with KPC and NDM Carbapenemase-Producing Enterobacterales”, the researchers compared the patients’ characteristics between the groups tested positive for KPC-CPE vs NDM-CPE. The researchers found characteristics such as admission origin and Charlson Comorbidity Index (CCI) were associated with the groups. While the research question is interesting, major improvements are needed.

Major comments:

  1. The univariate and multivariate regression analyses can only demonstrate associations between patients’ characteristics and KPC-CPE vs NDM-CPE colonization. AUCs or F1 scores on a test dataset are needed to demonstrate the predictive values of these characteristics. This is a major concern as predicting NDM-CPE patients is one conclusion from the paper.

Answer: We agree that multivariate regression analyses can only demonstrate associations between patients’ characteristics and KPC-CPE vs NDM-CPE colonization. We aimed to understand the epidemiology better and develop effective screening strategies for NDM-CPE on patients' hospital admission. We did not aim to define the predictive values of these characteristics, this is beyond the scope of out study.

  1. How do the characteristics distinguish between NDM-CPE vs non CPE patients?

Answer: This was beyond the scope of this research, therefor not mentioned in the manuscript. We published a prior article titled "Risk Factors and Outcome of Patients Colonized with Carbapenemase- Producing and Non-Carbapenemase-Producing Carbapenem-Resistant Enterobacteriaceae" in Infect Control Hosp Epidemiol. 2020;41:1154-1161. In that research we compared adult hospitalized patients with CP-CRE positive rectal swab cultures, non–CP-CRE positive rectal swab cultures, and negative rectal swab cultures (non-CRE). We found that patients with CP-CRE and non–CP-CRE versus no CRE more frequently resided in nursing homes (P<0.001), received antibiotics 3 months prior to admission (P < .001), and received glucocorticosteroids 3 months prior to admission (P = .047 and P < .001, respectively).

  1. Please clarify on which comorbidity these patients had and whether any of them were associated with the groups. Knowing the specific co-morbidities associated with NDM-CPE can be more informative than CCI.

Answer: We did not find a specific co-morbidity which was more prevalent in CPE-NDM colonized patients, we did find those that were more prevalent in KPC-NDM colonized patients (page 2, lines 84-86). We added a supplementary table (sTable 1), page 9.

  1. Since the characteristics were added stepwise to the multivariate regression model, what was the order of adding the characteristics? How were the different multivariate regression models compared and which model was determined as the final model?

Answer: Step 1 was omnibus test of model coefficients with the following variables added in the equation (in the following order): admission from home, age, CCI<4, solid tumors, urinary catheter, and decubitus ulcer. Age and solid tumors were not found as independent risk factors predicting NDM-CPE colonization. 

  1. From Table 3, all the variables were associated with the groups (p<0.05) except Age < 65, unless a different p value threshold was used. If a different threshold was used, please clarify, if not, why the other variables (urinary catheter and decubitus ulcer) were not mentioned and discussed?

Answer: We added these findings to the results section (page 4, line 113) and in the discussion section (page 4, line 135-136).

Minor comments:

  1. I would suggest changing “sig.” in Table 3 to “p-value”, same as in Table 1&2.

Answer: "sig." in Table 3 was changed to “p-value”.

  1. Although explained in the results section, I would suggest to also add footnotes in the tables to clarify the direction of the odds ratios between the groups.

Answer: Thank you for this comment, we rewrote table 3 in regard to the OR and 95% CI which we think now is clearer and therefor did not add footnotes (page 4). If you think they are still necessary, we will add them.

Reviewer 3 Report

Comments and Suggestions for Authors

Dear authors,

I found very interesting your colonization based study. Only a few comments

1. In what kind of population is SOROKA University Medical center refer to?

2. Patients carrying KPC-CPE were not detected at all, after June 2018, were they?

3. Patients coming from home should be divided into two groups, those having recurrent admissions, and those not having, because CPE colonization lasts more than a year. You should comment on this.

4. Was patients' mortality due to CPE infection?

5. What was the percentage of CPE carriers that developed infection?

Author Response

Answers to Reviewer 3

Reviewer 3: Dear authors, I found very interesting your colonization based study. Only a few comments

  1. In what kind of population is SOROKA University Medical center refer to?

Answer: Soroka University Medical Center serves as a referral center for Southern Israel's population which is comprised of patients of Jewish and Bedouin-Arab ethnicity (75% and 25%, respectively). We added this to the study design, page 5, lines 186-187.

  1. Patients carrying KPC-CPE were not detected at all, after June 2018, were they?

Answer: Patients colonized with KPC-CPE were detected after June 2018, though KPC-CPE has decreased from 73% to less than 30% of all CPE isolates while NDM-CPE increased from 9% to 60%. We chose to use a historical cohort of KPC-CPE colonized patients which we previously studied (reference #14).

  1. Patients coming from home should be divided into two groups, those having recurrent admissions, and those not having, because CPE colonization lasts more than a year. You should comment on this.

Answer:  Within NDM-CPE colonized patients, 110/136 (80.9%) were hospitalized from home, of whom 79 (71.8%) were hospitalized in the previous 6 months. Within KPC-CPE colonized patients, 29/92 (80.9%) were hospitalized from home, of whom 18 (62%) were hospitalized in the previous 6 months. This was added to the results section, page 2, lines 86-87.

We included in the study the 1st positive rectal CPE screening culture per patient. We might have missed CPE colonization in a previous hospitalization if the patients didn't comply with our current local and national screening criteria, one of them being recurrent admission, therefor found only on his/her second admission. This leads to one of the main reasons we conducted the study. We are of the opinion that our current local and national CPE screening criteria pertain to KPC-CPE but not fully to NDM-CPE.

  1. Was patients' mortality due to CPE infection?

Answer: Our patient population were colonized with CPE but not infected with CPE during their hospitalization. Many patients died after hospital discharge, we do not have data regarding their cause of death.

  1. What was the percentage of CPE carriers that developed infection?

Answer: At the time of the 1st positive CPE screening culture results, none of the patients had previous positive CPE clinical culture results. During the current hospitalization off the 1st positive CPE screening culture result, 9 /92 (9.8%) patients colonized with KPC-CPE developed an infection and 10 /136 patients (7.3%) colonized with NDM-CPE developed an infection (2 cases NDM-CPE bacteremia and 1 case of KPC-CPE bacteriuria, the rest were mainly urinary tract infections).

Round 2

Reviewer 2 Report

Comments and Suggestions for Authors

All the comments were comprehensively addressed.